# *KIF6* Trp719Arg Genetic Variant Increases Risk for Thoracic Aortic Dissection

**DOI:** 10.3390/genes14020252

**Published:** 2023-01-18

**Authors:** Juan J. Velasco, Yupeng Li, Bulat A. Ziganshin, Mohammad A. Zafar, John A. Rizzo, Deqiong Ma, Hui Zang, Asanish Kalyanasundaram, John A. Elefteriades

**Affiliations:** 1Aortic Institute, Yale University School of Medicine, New Haven, CT 06510, USA; 2Department of Statistics, Rowan University, Glassboro, NJ 08028, USA; 3Department of Statistics, Stony Brook University, Stony Brook, NY 11794, USA; 4DNA Diagnostics Laboratory, Department of Genetics, Yale University School of Medicine, New Haven, CT 06510, USA

**Keywords:** aortic dissection, *KIF6*, kinesin, aortic aneurysm, ascending aorta, risk prediction

## Abstract

Background: KIF6 (kinesin family member 6), a protein coded by the *KIF6* gene, serves an important intracellular function to transport organelles along microtubules. In a pilot study, we found that a common *KIF6* Trp719Arg variant increased the propensity of thoracic aortic aneurysms (TAA) to suffer dissection (AD). The present study aims for a definite investigation of the predictive ability of *KIF6* 719Arg vis à vis AD. Confirmatory findings would enhance natural history prediction in TAA. Methods: 1108 subjects (899 aneurysm and 209 dissection patients) had *KIF6* 719Arg variant status determined. Results: The 719Arg variant in the *KIF6* gene correlated strongly with occurrence of AD. Specifically, *KIF6* 719Arg positivity (homozygous or heterozygous) was substantially more common in dissectors (69.8%) than non-dissectors (58.5%) (*p* = 0.003). Odds ratios (OR) for suffering aortic dissection ranged from 1.77 to 1.94 for Arg carriers in various dissection categories. These high OR associations were noted for both ascending and descending aneurysms and for homozygous and heterozygous Arg variant patients. The rate of aortic dissection over time was significantly higher for carriers of the Arg allele (*p* = 0.004). Additionally, Arg allele carriers were more likely to reach the combined endpoint of dissection or death (*p* = 0.03). Conclusions: We demonstrate the marked adverse impact of the 719Arg variant of the *KIF6* gene on the likelihood that a TAA patient will suffer aortic dissection. Clinical assessment of the variant status of this molecularly important gene may provide a valuable “non-size” criterion to enhance surgical decision making above and beyond the currently used metric of aortic size (diameter).

## 1. Introduction 

KIF6 (Kinesin family member 6) is a protein encoded by the *KIF6* gene. Proteins in this family serve a vital function as microtubule motors that transport vesicles, organelles, protein complexes, and messenger ribonucleic acids toward the cell nucleus [1,2]. Typical kinesins are homodimeric molecules consisting of two N-terminal domains (“heads”) that move along microtubles and C-terminal domains (“tails”) that interact directly with the transported cargos or indirectly through adapter molecules. The *KIF6* 719Arg polymorphism (NM_145027.6(*KIF6*):c.2155T > C (*p*.Trp719Arg), 6:39325078-A g (hg19), rs20455) replaces a non-polar residue (Trp) with a basic residue (Arg) near the cargo-binding tail domain; thus, this polymorphism has the potential to alter the affinity of the cargo molecules or possibly to modulate the motor activity of the KIF6 protein. 

Extensive investigations over a decade (including large-scale prospective studies) identified a close association between one specific *KIF6* variant (*KIF6* 719Arg) and the incidence of myocardial infarction. Additionally, in extensive population studies, it was found that *KIF6* variants were associated with salutary reduction in myocardial infarction and fatal coronary events with statin therapy [1,2,3,4,5,6,7,8,9,10,11,12].

In a preliminary investigation [13], we were able to identify an association between the same *KIF6* variant and the likelihood of thoracic aortic aneurysm patients suffering an aortic dissection. Specifically, carriers of the *KIF6* 719Arg variant were found to have a two-fold increase in susceptibility to aortic dissection (OR 2.14, confidence interval (CI) 1.18–3.9). That study was based on 140 thoracic aortic dissection cases and 497 non-dissection cases from three countries (United States, Hungary, and Greece). 

In the present study, we have augmented the patient number substantially and have investigated once again the association between the *KIF6* 719Arg variant and the incidence of aortic dissection. The patient number is now large enough to permit sound conclusions, in contradistinction to the initial exploratory investigation previously performed. 

Confirmation of a close association between this specific variant in this gene of key importance and intracellular function could potentially improve natural history prognostication in thoracic aortic aneurysm (TAA) disease and enhance criteria for surgical intervention. A biomarker capable of enhancing prediction in TAA disease could help to prevent dreaded dissection events, while concurrently sparing patients from potentially unnecessary major thoracic aortic surgery. 

## 2. Materials and Methods

This study was approved by the Human Investigations Committee of Yale University (ID: 0109012617, ID1609018416) (and for the early specimens by the respective Institutional Review Boards of Semmelweis University in Budapest, Hungary and Evangelismos General Hospital in Athens, Athina, Greece).

Study subjects (including those from the original study and those recruited in intervening years) numbered 1108:899 with aortic aneurysm alone (henceforth called “aneurysm” patients), and 209 with aortic dissection in the setting of aneurysm disease (henceforth called “dissectors”). 

*KIF6* 719Arg variant determination was performed on blood specimens, by polymerase chain reaction (PCR) [5] by Celera Diagnostics (637 patients), by PCR by the Yale University Clinical Chemistry Laboratory or Quest Diagnostics (KIF6 Genotype, CardioIQ^®^ Quest Diagnostics, Secaucus, NJ, USA) (41 patients), or by whole exome sequencing of (430 patients) by the Yale University Genetics Laboratory.

We captured and analyzed the impact of *KIF6* variant status on three subtypes of dissection-spectrum pathologies: typical flap-type aortic dissection (henceforth called “dissection”), intramural hematoma (IMH), and penetrating aortic ulcer (PAU) [14]. These subtypes of dissection pathology were analyzed separately and then in combinations. 

For years, we have applied a surgical threshold diameter of ~5.0 cm ascending aortic diameter, based on our natural history studies [15]. Many patients in this study underwent surgery when their aortic diameter exceeded 5 cm. This criterion has just been adopted in the latest societal guidelines [16]. So, our traditional surgical threshold has long been in accordance with the very newest guidelines. 

Positive family history refers to 1st order relatives, and such history was determined by the senior investigator in family consultation. 

Statistical analysis. Categorical variables were expressed as frequencies and percentages and were analyzed with the chi-squared test or the Fisher’s exact test, as appropriate. Normality was evaluated with the Komolgorov-Smirnov test. Continuous variables were normally distributed and were presented as means with standard deviations. Independent sample *t*-tests were used to analyze normal distributions. Univariable logistic regression models tested the association between the KIF6 genotypes and aortic dissection. Additionally, multivariable logistic regression models evaluated this association--including age, sex, hypertension, smoking status, bicuspid aortic valve, and presence of syndromic connective tissue disease as covariates. Odds ratios and their 95% confidence intervals were calculated. The Kaplan–Meier survival curves and the log-rank test or Gehan–Breslow test, as appropriate, were used to describe the effect of the *KIF6* 719Arg variant on freedom from aortic dissection and freedom from dissection or death. A two-tailed α level was set at 0.05. All statistical analyses were performed with R software, version 4.1.2 (R Foundation for Statistical Computing, Vienna, Austria).

## 3. Results

Characteristics of aneurysm and dissection patients are presented in Table 1. *KIF6* 719Arg status was ascertained in 1108 patients; there were 209 patients with thoracic aortic dissection and 899 patients with non-dissected thoracic aorta aneurysm. Of the 209 dissection cases, 181 were typical, 25 were IMH, and three were PAU. The mean age in the dissection group was 61.7 years, and in the non-dissected group, 60.8 years. The mean aortic size was 5.6 cm in the dissection group and 5.1 cm in the non-dissection group. A proportion of 16.2% of dissectors had a family history of aneurysm disease, and 24.2% of non-dissectors. The Caucasian race predominated in both dissectors (91.3%) and non-dissectors (97.8%). The dissectors had a higher percentage of descending and combined ascending/descending aneurysms (55.4%) than the non-dissectors (15.6%). Bicuspid valves were more common in the non-dissectors (1.9% vs. 15.2%). Hypertension was more common in the dissectors (76% vs. 65.8%). 

Table 2 indicates the relationship between the presence or absence of dissection and *KIF6* 719Arg status. Please note that having any Arg allele (homozygous or heterozygous, that is Arg/Arg or Arg/Trp) was significantly more common in dissectors (69.8%) than in non-dissectors (58.5%) (*p* = 0.003). Even just being heterozygous for a single Arg allele also was highly significantly more common in the dissectors (51.6%) than the non-dissectors (44.4%) (*p* = 0.008).

Table 3 presents the data from the converse point of view: that is, clinical characteristics by *KIF6* 719Arg allele category. Please note that all *KIF6* 719Arg categories (homozygous or heterozygous) had a statistically higher fraction of dissectors than the non-Arg category (*p* = 0.003).

Table 4 presents the odds ratios for *KIF6* 719Arg positivity for patients with each aortic dissection type. We separated by typical, IMH, and PAU characteristics of the dissection phenomenon. Please note the substantially higher odds ratios of *KIF6* 719Arg positivity for all combinations of categories, ranging from 1.63 to 1.94. The OR for typical dissection (probably the most clinically relevant and discrete) ranges from 1.63 for the hetero- or homozygous category to 1.94 for the exclusively homozygous category. These odds ratios were adjusted for age, sex, hypertension, smoking status, bicuspid aortic valve, and presence of syndromic connective tissue disease. 

Table 5 presents a brief summary of odds ratios for different dissection type groupings (typical, IMH, PAU)—all substantially elevated and statistically significant. 

Appendiceal Table A1, Table A2, Table A3, Table A4, Table A5 and Table A6 present data for various groupings of aortic dissection types: Either all (typical + IMH + PAU) or just (typical + IMH), each considered in cohorts with vs. without coronary artery disease (CAD)), and with ascending vs. descending aneurysms. Findings remain robust for all except isolated heterozygous descending categories.

Because concomitant CAD could potentially confound the observed association between the *KIF6* 719Arg variant and thoracic aortic dissection, we reanalyzed the data after excluding those with CAD. The adjusted OR ratio remained significant after that adjustment (OR 1.74). We performed this additional analysis for completeness, despite the fact that our extensive clinical aortic investigations have shown that ascending aortic aneurysm patients (in contradistinction to descending and thoracoabdominal patients) are highly protected from atherosclerosis, as assessed by multiple accepted measures, including intimal medial thickness (IMT), total calcium score, and rate of coronary artery disease and myocardial infarction [17,18,19].

Figure 1 shows survival free of aortic dissection (starting at 20 years of age; dissections are very rare in the teenage years). Please note that the two curves diverge, with a higher incidence of dissection in the *KIF6* 719Arg carrier group (*p* = 0.04). Figure A1 in the appendix shows a small but significant increase in the rate of dissection or death in the *KIF6* 719Arg carrier group. 

In an effort to minimize the possibility of any confounding factors, we performed a number of additional statistical calculations. (1) Analysis of Caucasian patients only. To counteract influence of ethnic factors, we recalculated the ORs with all non-Caucasian patients eliminated. The ORs continued to be robust and strongly significant statistically. (2) Analysis after exclusion of patients with family history of aortic disease. To counteract the influence of any family clusters within the patients studied, we eliminated the seven patients who were enrolled from families with more than one member represented in our study. The ORs continued to be robust and strongly significant statistically. (3) Exclusion of batch effect. To exclude possible batch effect from the original vs. the supplemental cohort, we performed statistical analysis on the new patient cohort (471 patients) separately. The ORs continued to be robust and strongly significant statistically. 

Finally, the *KIF6* 719Arg variant that we are investigating is known to appear in cis with another *KIF6* variant (6:39325077-C-T), albeit an extremely rare one (gnomAD frequency of 5.3 × 10^−5^ in the European non-Finish cohort). The combined effect of this multiallelic variant is to change the tryptophan residue to glutamine. In analyzing our exome sequencing cohort of patients (*n* = 430) to determine the presence of this multiallelic *KIF6* variant, we did not find it to be present in any of our patients.

Serial aortic measurements were available in 102 patients in this study. Mean growth rate was 0.05 cm/year in *KIF6* positive patients and 0.01 cm/year in *KIF6* negative patients. Although this data indicates more rapid growth in the *KIF6* variant patients, this result did not reach statistical significance. We look forward to more robust growth rate data (as more patients are studied in the future) in order to determine if the noted trend toward more rapid growth in the *KIF6* variant patients might be confirmed statistically.

Table 6 presents the results of whole exome sequencing performed in the patients in this study. We see many aneurysm genes represented in both *KIF6* variant carriers and non-carriers (at various levels of pathogenetic likelihood). 

## 4. Discussion

We have found, in this large sample of thoracic aortic aneurysm patients, that the *KIF6* 719Arg genetic variant confers substantially increased risk of aortic dissection. The ORs approaching 2.00 are quite substantial for association of a single genetic polymorphism with a complex clinical disease. This risk was reflected in a statistically significantly higher rate of dissection over the long term as shown by Kaplan–Meier analysis. Aortic dissection, especially when presenting lethally, is often underdiagnosed. Accordingly, we also assessed the combined endpoint of dissection (diagnosed) or death; for this metric as well, *KIF6* 719Arg positivity conferred a statistically significant increased risk. 

It is worth examining the biological plausibility of an important clinical impact of *KIF6* on aortic biology and clinical prognosis. Firstly, KIF6 has been localized in blood vessels and in the endothelium [20]. Furthermore, another study found, in zebrafish, that mutations in *KIF6* are related to scoliosis [21]. We know that skeletal abnormalities are prominent in syndromic thoracic aortic diseases (e.g., Marfan disease). Furthermore, very recently published research from our own team [22] has found that scoliosis appears to be part of the zebrafish phenotype of variants that cause thoracic aortic aneurysm in humans. Thus, there appears to be fundamental biological evidence supporting a possible role of the kinesin family, and KIF6 in particular, in promoting aortic aneurysm disease. 

We have briefly described in the introduction the important intracellular roles served by kinesins such as KIF6 and the early studies by Iakoubova, Schiffman, and colleagues [2,3,4,5,6,7,8,9,10,11,12], demonstrating that KIF6 is intimately involved with atherosclerosis and also with statin responsiveness. The fundamental molecular roles of KIF6 and its clinical associations are certainly critical enough that a major physiologic impact would not be unexpected from variations in its genetics. In terms of any specific potential mechanisms for predisposition to aortic dissection, these are currently unknown. Additionally, although original studies have associated *KIF6* variants with coronary artery disease (CAD) and statin responsiveness, this connection has been questioned as more data have become available [23] In our study (Table A1) we find a lack of association between CAD and dissection. This dissociation between CAD and ascending aortic disease is in keeping with multiple studies from our team [17,18,19] showing that ascending aortic aneurysm patients are remarkably free of atherosclerotic disease; in fact, ascending aortic aneurysm patients have less total body vascular calcification, a lower intimal medial thickness (IMT) in the carotid artery, and almost complete protection from myocardial infarction. Additionally, ascending aortic aneurysm patients have much lower LDL than non-aneurysmal patients [24] These findings seem paradoxical compared to general dogma regarding atherosclerosis until one recognizes that ascending aneurysms are non-calcified, smooth contoured, and non-thrombus containing. This stands in contradistinction to descending and abdominal aortic aneurysms, which are frankly atherosclerotic, with heavily calcified walls, irregular contour, and heavy thrombus burden. These novel perspectives differentiating ascending from descending aortic disease are also completely consonant with the findings in Table A2 that *KIF6* has as a stronger impact on ascending than on descending aortas.

Our dissected aortas were, on average, 0.52 cm larger than the non-dissected aortas; this is fully consistent with the findings by our group and others that the aorta grows substantially at the moment of aortic dissection [15,25]

Limitations: Despite the markedly expanded patient pool, considerable for a rather infrequently diagnosed disease like aortic dissection, this study has limitations.

We have demonstrated important associations of the *KIF6* 719Arg variant with aortic dissection, which we hope will enhance prediction and patient triage. However, this association does not prove causation. The protean, fundamental actions of KIF6 suggest that a role in causation is indeed plausible, although this awaits elucidation.The *KIF6* 719Arg variant is very common in the general population (0.41 overall frequency as shown in gnomAD and other genetic databases) [26], arguing against a strong physiologic effect from this gene alone. We suspect that the adverse effect of *KIF6* is minimal unless combined synergistically with other disease-causing aneurysm variants (such as those in Table 6), in which case *KIF6* 719Arg increases the overall virulence. Nonetheless, within the limited context of our thoracic aortic patient group, *KIF6* 719arg variant status appears to confer a substantial superimposed vulnerability to aortic dissection. We see *KIF6* 719Arg not as a conventional disease-causing sole variant, but as contributor to dissection outcome which can serve as a predictor to encourage increased clinical monitoring of aneurysm patients and, possibly influence the clinical decision to operate.Patients dying upon initial presentation from aortic dissection are generally not included in our *KIF6* analysis, as *KIF6* 719Arg testing, a specialized laboratory test, would not have been a priority or a feasibility under such urgent circumstances. The impact of absent lethal cases on the analysis could be substantial [27]. For example, a presumptive preferential mortality of Arg/Arg homozygotes could artificially underestimate the predictive effect of *KIF6*. If anything, this factor makes our positive findings more rather than less cogent.The determination of *KIF6* 719Arg variant status was done via multiple means. However, since *KIF6* 719Arg variant status is a binary characteristic, it is unlikely to be influenced by the specific method of determination. A quantitative, non-binary characteristic would be more susceptible to such an issue.Midway through this study, we began to recognize a “predictive” ability of the *KIF6* 719Arg variant vis à vis aortic dissection. Accordingly, we triaged a number of patients to surgery partly on this basis. This factor would also tend artificially to underestimate the effect of *KIF6*. If anything, this factor makes our positive findings more rather than less cogent.Additionally, although we included patients from three different geographic sources, there were few non-Caucasian participants. So, the findings may not be representative for individuals of all origins.Our study did not include gene expression analysis, which may, in future studies, clarify the biological pathways through which the *KIF6*’s impact is exerted.It is possible that any non-dissectors in this study who were operated at low surgical threshold diameters could have progressed to dissection if followed for longer periods. However, recent evidence from our group and from others [15] has shown that the aorta expands acutely at the moment of aortic dissection—by ~0.8 cm. So, this would bring the true pre-dissection size of our dissectors down below the size of our non-dissectors, likely abrogating this concern.We consider our findings to me most relevant to typical aortic dissection (the most common type by far), as intramural hematoma and penetrating aortic ulcer, while often included in the category of dissection phenomena, are very different clinical entities.

Despite these limitations, we are encouraged by the robustness of our *KIF6* 719Arg findings along two axes. Firstly, the findings achieve high odds ratios and levels of statistical significance, unusual for a single biological variable in a complex disease. Secondly, the findings remain robust no matter which way the patients are grouped—ascending or descending or combined—or typical dissection with or without IMH and PAU. [28]

We feel that this study raises the possibility that *KIF6* 719Arg variant status may serve as a useful clinical predictor for the likelihood of aortic dissection. If so, this *KIF6* 719Arg status would represent a welcome “non-size” criterion for surgical intervention, broadening the scope of variables to be considered in clinical decision making. In a patient with borderline aortic enlargement—a so-called “judgment call” case—*KIF6* 719Arg positivity might reasonably tilt the decision in favor of prophylactic aortic surgery. The advent of such “non-size” criteria promises to enhance decision making regarding surgical intervention in aortic disease, along avenues not previously available. 

## 5. Conclusions

In our initial pilot study of *KIF6* 719Arg’s impact on aortic dissection, we stated: “If the association of the *KIF6* 719Arg variant with thoracic aortic dissection is further confirmed, this variant could be useful in assessing thoracic aortic dissection risk”. The present study provides the further confirmation that was eagerly anticipated. 

While we are excited to present this interesting data regarding another clinical factor to inform decision making for aneurysm patients, we hasten to point out that we consider our findings to be valuable yet still emerging clinical data, worthy of further replication and confirmation.

## Figures and Tables

**Figure 1 genes-14-00252-f001:**
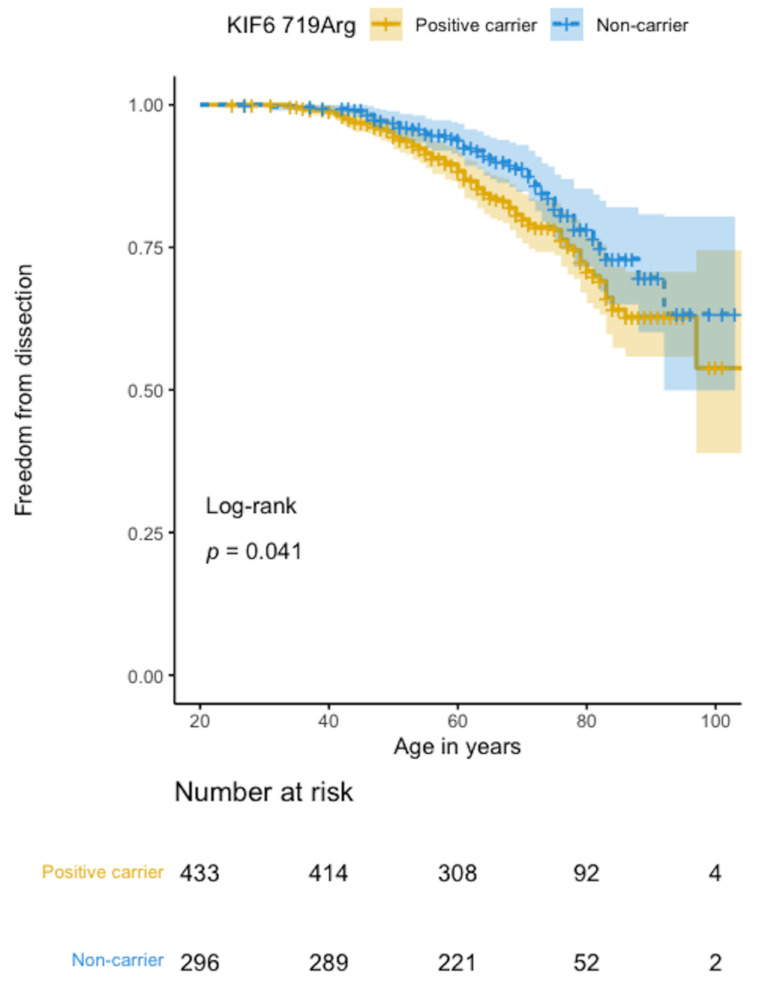
Kaplan–Meier estimate of freedom from aortic dissection. Note the statistically significantly increased rate of events in the positive carriers.

**Table 1 genes-14-00252-t001:** Patient characteristics.

	Dissectors *	Aneurysm (Non-Dissected)	*p* Value
*n*	209	899	
Age (mean, SD)	61.7 (13.5)	60.8 (13.8)	0.020
Size (mean, SD)	5.6 (1.5)	5.13 (1)	0.008
Female (%)	80 (38.2)	253 (28.1)	0.005
Location of aneurysm (%)			<0.001
Ascending	91 (43.5)	758 (84.3)	
Descending	91 (43.5)	112 (12.4)	
Ascending and descending	25 (11.9)	29 (3.2)	
Unknown	2 (0.9)	0 (0)	
Rupture (%)	7 (3.3)	8 (0.8)	0.012
Family history (%)	34 (16.2)	218 (24.2)	0.016
Bicuspid aortic valve (%)	4 (1.9)	137 (15.2)	<0.001
Center (%)			0.221
United States	169 (80.8)	749 (83.3)	
Hungary	28 (13.3)	86 (9.5)	
Greece	12 (5.7)	64 (7.1)	
Race (%)			<0.001
Caucasian	191 (91.3)	880 (97.8)	
Black or African American	12 (5.7)	3 (0.3)	
Asian	3 (1.4)	5 (0.5)	
Hispanic	1 (0.4)	8 (0.8)	
Indian American or Alaska Native	1 (0.4)	0 (0)	
Indian	0 (0)	1 (0.1)	
Other	1 (0.4)	2 (0.2)	
Hypertension (%)	159 (76)	592 (65.8)	0.005
Smoking status (%)	89 (42.5)	247 (27.4)	<0.001
Connective tissue disease (%)	9 (4.3)	9 (1)	<0.001

***** Typical dissection, intramural hematoma, and penetrating atherosclerotic ulcer cases. *n* = number.

**Table 2 genes-14-00252-t002:** *KIF6* genotypes, by dissectors vs. non-dissectors.

	Dissectors *	Aneurysm (Non-Dissected)	*p* Value ^◊^
*n*	209	899	
Genotype (%)			
Arg/Arg + Arg/Trp	146 (69.8)	526 (58.5)	0.003
Arg/Arg	38 (18.1)	126 (14)	0.015
Arg/Trp	108 (51.6)	400 (44.4)	0.008
Trp/Trp	63 (30.1)	373 (41.4)	

**^◊^** The *p* value was based on a comparison with non-carriers (Trp/Trp). ***** Typical dissection, intramural hematoma, and penetrating atherosclerotic ulcer cases. *n* = number.

**Table 3 genes-14-00252-t003:** Likelihood of dissection (and aneurysm details), by genotype.

	Arg/Arg + Arg/Trp	Arg/Arg	Arg/Trp	Trp/Trp	*p* Value ^◊^
*n*	672	164	508	436	
Age (mean, SD)	60.5 (14)	61 (13.9)	60.4 (14.1)	61.6 (13.2)	0.180
Dissectors * (%)	146 (21.7)	38 (23.1)	108 (21.2)	63 (14.4)	0.003
Female (%)	216 (32.1)	58 (35.3)	158 (31.1)	117 (26.8)	0.069
Location of aneurysm (%)					0.015
Ascending	495 (73.6)	112 (68.2)	383 (75.3)	354 (81.1)	
Descending	139 (20.6)	41 (25)	97 (19)	64 (14.6)	
Ascending and Descending	37 (5.5)	11 (6.7)	27 (5.3)	17 (3.8)	
Unknown	1 (0.1)	0 (0)	1 (0.1)	1 (0.2)	

**^◊^** The *p* values were based on a comparison of positive carriers (Arg/Arg + Arg/Trp) with non-carriers (Trp/Trp); ***** Typical dissection, intramural hematoma, and penetrating atherosclerotic ulcer cases. *n* = number.

**Table 4 genes-14-00252-t004:** Association of *KIF6* 719Arg genotype with thoracic aortic dissection categories. (See Reference 19 for detailed discussion of typical and variant (IMH/PAU) dissection pathologies.).

		Unadjusted			Adjusted *		
Diagnostic Grouping	*n*	OR	95% CI	*p* Value	OR	95% CI	*p* Value
Dissection (typical only)							
Arg/Arg + Arg/Trp	128	1.69	1.20–2.41	0.002	1.71	1.20–2.46	0.003
Arg/Arg	36	2.02	1.26–3.22	0.003	1.94	1.19–3.15	0.007
Arg/Trp	92	1.59	1.11–2.31	0.012	1.63	1.12–2.39	0.010
Trp/Trp	53	REF			REF		
Dissection (Typical and IMH)							
Arg/Arg + Arg/Trp	146	1.73	1.25–2.43	<0.001	1.74	1.25–2.47	0.013
Arg/Arg	38	1.88	1.19–2.95	0.006	1.77	1.09–2.83	0.017
Arg/Trp	108	1.69	1.20–2.40	0.002	1.73	0.84–1.93	0.002
Trp/Trp	60	REF			REF		
Dissection (All-Typical, IMH, PAU)							
Arg/Arg + Arg/Trp	146	1.62	1.19–2.82	0.002	1.64	1.17–2.31	0.003
Arg/Arg	38	1.78	1.12–2.78	0.012	1.66	1.03–2.65	0.033
Arg/Trp	108	1.59	1.13–2.25	0.007	1.63	1.14–2.33	0.006
Trp/Trp	63	REF			REF		

***** Adjusted to age, sex, family history, bicuspid aortic valve, hypertension, smoking history, and connective tissue disease; IMH, intramural hematoma; PAU, penetrating atherosclerotic ulcer; OR, odds ratio; CI, confidence interval; REF, reference. Connective tissue disease = Marfan disease, Ehlers–Danlos syndrome, or Loeys-Dietz syndrome. *n* = number.

**Table 5 genes-14-00252-t005:** Summary table. Impact of *KIF6* 719Arg positivity on probability of aortic dissection. Ranges under the odds ratio indicate the lowest and highest adjusted levels depending on exact genotype (Arg/Arg + Arg/Trp, Arg/Arg, or Arg/Trp). Extracted for clarity from Table 4.

Diagnostic Grouping	*n*	Odds Ratio	95% Confidence Interval
Dissection (all—typical, IMH, PAU)	146	1.63–1.66	1.03–2.65
Dissection (typical and IMH)	146	1.73–1.77	0.84–2.83
Dissection (typical only)	128	1.63–1.94	1.12–3.15

*n* = number.

**Table 6 genes-14-00252-t006:** Genetic variants found on whole exome sequencing of patients in this study. These are classified here according to the standard American College of Medical Genetics and Genomics categories.

	Pathogenic	Likely Pathogenic	VUS (Variants of Uncertain Significance)
*KIF6* variant carriers	5 cases: *TGFB3*, *MYLK*, *TGFBR2*, *SMAD3*, *FBN1*	5 cases: *TGFBR2*, *FBN1*	70 cases: *MIB1, TGFBR3, FBN1, MYH11, COL5A1, ACTA2, MYLK, MYH11, COL1A1, PRKG1, TGFB2, COL3A1, NOTCH1, FLNA, FBN2, SMAD3, COL5A2, TNXB, LOX, HCN4, EMILIN1, ELN, GATA5, BGN*
*KIF6* Variant non-carriers	6 cases: *FBN1*, Xp deletion, *PLOD1*, *SMAD3*	1 case: *TNXB*	49 cases: *COL5A2, TGFBR1, ACTA2, MYH11, FBN1, COL1A1, COL3A1, NOTCH1, FLNA, GATA4, FBN2, COL5A1, MIB1, FLNA, TGFBR2, FBN2, LOX, PRKG1, PKD2*

## Data Availability

The data presented in this study are available on request from the corresponding author. The data are not publicly available due to privacy concerns.

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
