# Peer review of "KIF6 Trp719Arg Genetic Variant Increases Risk for Thoracic Aortic Dissection"

_genes, 2023, doi:10.3390/genes14020252_

Round 1

Reviewer 1 Report

This is a very exciting paper with findings that are of great importance to the field of aortic surgery, and with further validation, these findings could positively impact decision-making processes such that aortic repairs are more appropriately undertaken. Maximum diameter has long been acknowledged to be an extremely poor indicator of when intervention is required, and a biomarker has the potential to be a genuine game-changer. 

Can the authors please clarify if the patients recruited in this study were all patients on whom aortic repair was undertaken? If this is the case, was consideration given to analysing the aortic tissue, including gene expression? 

I note that the mean diameter in non-dissectors was below the 5.5cm threshold for intervention. This makes one wonder if the threshold for intervention was higher, how many non-dissectors would have actually progressed to dissection. 

Considering the mean diameter is below the threshold, can the authors give some insight as to the reason for intervention, e.g. rapid expansion etc.? I see only 15.2% had bicuspid aortic valve, only 1% had connective tissue disorder, and only 0.8% had ruptured, so these were unlikely to have contributed to the decision to operate at a lower threshold.

Do a substantial proportion, if any, of the patients have longitudinal imaging data? If so, have the authors considered correlating the growth rate with the KIF6 Trp719Arg variant? Perhaps a longitudinal study of aortic patients under surveillance with cross-sectional imaging would be of interest to investigate how the presence of the KIF6 Trp719Arg variant correlates with aortic wall behaviour. 

I note from table 1 that there are a number of confounding factors that are significant predictors of adverse aortic events, which are almost universally higher in the dissectors group (with the exception of BAV and family history).  From appendix A, we can see that adjusting for these confounding factors has a variable influence on the significance of the genetic variants relative to whether the pathology is in the ascending or descending aorta, in particular, the Arg/Trp genotype. This needs to be teased out further in the discussion and is of particular interest considering the known differences in aortic wall composition, function and embryological origin between the ascending and descending aorta.

Can we please get a breakdown of the number of typical vs IMH vs PAU cases? i.e. are the numbers of IMH or PAU large enough to influence statistical outcomes? This will aid the interpretation of tables 4 and 5 and the tables in appendix A.

The presentation of dissection appears atypical in that there are equal numbers of ascending (type A) and Descending (type B). Type A dissections would usually present twice as frequently as type B. Can you explain this and also explain why type Bs were getting surgical intervention, if indeed there all were? Were the type B dissections acute, subacute, or chronic? 

A small typo on line 36, page 1: please remove the first "directly"

Author Response

Reviewer 1

We are extraordinarily grateful to both reviewers. Their extremely thoughtful and insightful comments and suggestions have led us to improve our manuscript substantially, to provide important additional data and clarifications, and to make plans for further extensions of this work in promising new directions.

This is a very exciting paper with findings that are of great importance to the field of aortic surgery, and with further validation, these findings could positively impact decision-making processes such that aortic repairs are more appropriately undertaken. Maximum diameter has long been acknowledged to be an extremely poor indicator of when intervention is required, and a biomarker has the potential to be a genuine game-changer. 

We thank the Reviewer for the generous comments, which encourage us in this work.  

Can the authors please clarify if the patients recruited in this study were all patients on whom aortic repair was undertaken? If this is the case, was consideration given to analysing the aortic tissue, including gene expression? 

This study included both surgical and non-surgical patients in about equal proportions.  We have performed a prior RNA gene expression study on a different group of patients. For that study, we sampled blood levels in our patients with thoracic aortic aneurysm. (Wang Y et. al. Gene expression signature in peripheral blood detects thoracic aortic aneurysm. PLoS One. 2007;17:2(10):e1050. PMID: 17940614 . DOI: 10.1371/journal.pone.0001050 .) In that study, we found that a 41-gene “classifier” based on expression signature was able to identify TAA patients with high accuracy. We did not perform any gene expression analysis in the current study. We thank the reviewer for pointing out the potential importance of gene expression studies in terms improving understanding of the KIF6 impact. We appreciate your deep insight that  such studies may identify the pathways through which KIF6 influence is exerted.  Based on the Reviewer’s insightful suggestion, we will plan to include such analysis in our next study. These gene expression analyses are expensive and will require dedicated funding.

We have added the following brief sentence to the Limitations section of the Discussion section of our manuscript.

Our study did not include gene expression analysis, which may, in future studies, clarify the biological pathways through which the KIF6 impact is exerted.

I note that the mean diameter in non-dissectors was below the 5.5cm threshold for intervention. This makes one wonder if the threshold for intervention was higher, how many non-dissectors would have actually progressed to dissection. 

We have for years utilized, and championed, a surgical threshold of 5.0 cm for good risk surgical candidates. This lower threshold was just recently adopted in the latest societal guidelines. We understand the Reviewer’s important point. Accordingly, we have added the following sentence to the Limitations section of the Discussion.

It is possible that any non-dissectors in this study who were operated at low surgical threshold diameters could have progressed to dissection if followed for longer periods. However, recent evidence from our group and from others (Ziganshin BA, Zafar MA, Elefteriades JA. Descending threshold for ascending aortic aneurysmectomy: Is it time for a "left-shift" in guidelines? J Thorac Cardiovasc Surg. 2019 Jan;157(1):37-42. doi: 10.1016/j.jtcvs.2018.07.114. Epub 2018 Oct 23. PMID: 30557953.) (Rylski B, Blanke P, Beyersdorf F, et al, How does the ascending aorta geometry change when it dissects? J Am Coll Cardiol. 2014;63:1311-1319) has shown that the aorta expands acutely at the moment of aortic dissection—by ~0.8 cm. So, this would bring the size of our dissectors down below the size of our non-dissectors, likely abrogating this concern.

Considering the mean diameter is below the threshold, can the authors give some insight as to the reason for intervention, e.g. rapid expansion etc.? I see only 15.2% had bicuspid aortic valve, only 1% had connective tissue disorder, and only 0.8% had ruptured, so these were unlikely to have contributed to the decision to operate at a lower threshold.

Please see response to item just above. We have added the following disclaimer to the Methods section.

For years, we have applied a surgical threshold diameter of ~5.0 cm ascending aortic diameter, based on our natural history studies. (Ziganshin BA, Zafar MA, Elefteriades JA. Descending threshold for ascending aortic aneurysmectomy: Is it time for a "left-shift" in guidelines? J Thorac Cardiovasc Surg. 2019 Jan;157(1):37-42. doi: 10.1016/j.jtcvs.2018.07.114. Epub 2018 Oct 23. PMID: 30557953.) (Rylski B, Blanke P, Beyersdorf F, et al, How does the ascending aorta geometry change when it dissects? J Am Coll Cardiol. 2014;63:1311-1319) Many patients in this study underwent surgery when their aortic diameter exceeded 5 cm. This criterion has just been adopted in the latest societal guidelines. So, our surgical threshold has traditionally been in accordance with the very newest guidelines.

Do a substantial proportion, if any, of the patients have longitudinal imaging data? If so, have the authors considered correlating the growth rate with the KIF6 Trp719Arg variant? Perhaps a longitudinal study of aortic patients under surveillance with cross-sectional imaging would be of interest to investigate how the presence of the KIF6 Trp719Arg variant correlates with aortic wall behaviour. 

Thank you for urging us to perform this very important analysis. We have added the following sentences to the Results section.

Serial aortic measurements were available in 102 patients in this study. Mean growth rate was 0.05 in KIF6 positive patients and 0.01 in KIF6 negative patients. Although this data indicates more rapid growth in the KIF6 variant patients, this result did not reach statistical significance. We look forward to more robust growth rate data (as more patients are studied in the future) in order to determine if the noted trend toward more rapid growth in the KIF6 variant patients might be confirmed statistically.

I note from table 1 that there are a number of confounding factors that are significant predictors of adverse aortic events, which are almost universally higher in the dissectors group (with the exception of BAV and family history).  From appendix A, we can see that adjusting for these confounding factors has a variable influence on the significance of the genetic variants relative to whether the pathology is in the ascending or descending aorta, in particular, the Arg/Trp genotype. This needs to be teased out further in the discussion and is of particular interest considering the known differences in aortic wall composition, function and embryological origin between the ascending and descending aorta.

Thank you for this important observation. Reviewer 2 posed a similar question. We have now addressed this ascending/descending issue in the Discussion as follows, calling on our extensive prior studies of the fundamental differences in these two very different “aortas” (ascending and descending) to which you call our attention. We appreciate your asking us to clarify these points

Also, although original studies have associated KIF6 variants with coronary artery disease (CAD), this connection has been questioned as more data became available. (Topal EJ, Damani SB. The KIF6 Collapse. JACC. 2010;56:15640-1566.) In our study (Table A1) we find a lack of association between CAD and dissection. This dissociation between CAD and ascending aortic disease is in keeping with multiple studies from our team (REFS immediately below) showing that ascending aortic aneurysm patients are remarkably free of indications of atherosclerotic disease; in fact, ascending aortic aneurysm patients have less total body vascular calcification, a lower intimal medial thickness (IMT) in the carotid artery, and almost complete protection from myocardial infarction. Also, ascending aortic aneurysm patients have lower LDL than non-aneurysmal patients. These findings seem paradoxical compared to general dogma regarding atherosclerosis until one recognizes that ascending aneurysms are non-calcified, smooth contoured, and non-thrombus containing. This stands in contradistinction to descending and abdominal aortic aneurysms, which are frankly atherosclerotic, with heavily calcified walls, irregular contour, and heavy thrombus burden. These novel perspectives differentiating ascending from descending aortic disease are also completely consonant with the findings in Table A2 that KIF6 has as stronger impact on ascending than on descending aortas.

--Achneck H, Modi B, Shaw C, Rizzo J, Albornoz G, Fusco D, Elefteriades JA. Ascending thoracic aortic aneurysms are associated with decreased systemic atherosclerosis. Chest. 2005;128:1580-1586)

 --Hung A, Zafar M, Mukherjee S, Tranquilli M, Scoutt LM, Elefteriades JA: Carotid intima-media thickness provides evidence that as-cending aortic aneurysm protects against systemic atherosclerosis. Cardiology 2012; 123:71-77.

--Chau K, Elefteriades JA. Ascending thoracic aortic aneurysms protect against myocardial infarctions. Int J Angiol. 2014 Sep;23(3):177-82. doi: 10.1055/s-0034-1382288. PMID: 25317029; PMCID: PMC4169096.

--Weininger G, Ostberg N, Shang M, Zafar M, Ziganshin BA, Liu S, Erben Y, Elefteriades JA. Lipid proviles help to explain protection from systemic atherosclerosis in patients with ascending aortic aneurysm. J Thorac Cardiothorac Surg. 2022;163:e129-32. https://doi.org/10.1016/j.jtcvs.2021.09.031.

Can we please get a breakdown of the number of typical vs IMH vs PAU cases? i.e. are the numbers of IMH or PAU large enough to influence statistical outcomes? This will aid the interpretation of tables 4 and 5 and the tables in appendix A.

We have added the following information to the Results section:

Of the 209 dissection cases, 181 were typical, 25 were IMH, and 3 were PAU.

The presentation of dissection appears atypical in that there are equal numbers of ascending (type A) and Descending (type B). Type A dissections would usually present twice as frequently as type B. Can you explain this and also explain why type Bs were getting surgical intervention, if indeed there all were? Were the type B dissections acute, subacute, or chronic? 

Your point is very well taken. We have been seeing more Type B dissections than previously expected. The Type B’s were generally not receiving surgical intervention. We only operate for truly complicated Type B dissections, which are uncommon. (Charilaou P, Ziganshin BA, Peterss S, Rajbanshi BG, Rajakaruna C, Zaza KJ, Salloum MN, Mukherjee A, Tranquilli M, Rizzo JA, Elefteriades JA. Current Experience With Acute Type B Aortic Dissection: Validity of the Complication-Specific Approach in the Present Era. Ann Thorac Surg. 2016 Mar;101(3):936-43. doi: 10.1016/j.athoracsur.2015.08.074. Epub 2015 Oct 27. PMID: 26518373).

A small typo on line 36, page 1: please remove the first "directly"

Thank you for calling this to attention. Corrected.

Reviewer 2 Report

The efforts of the authors to broaden our insights into the complex genetic background of aortic aneurysms and dissection are very welcome and much needed. In particular this tool that can predict which patients may develop an dissection has important implication because it allows to select patients for early treatment with better outcome.  In this perspective the current study present a step forwards in the direction of personalized medicine for aortic disease. The manuscript is very well written and easy to read and understand.

As we all know association studies generate questions. Here about the biological effect of identified specific polymorphism in KIF6 on the aortic wall, especially in times were major molecular mechanism of aneurysm are being unraveled.

The authors are therefore encouraged to provide a bit more explanation on the link between the KIF6 polymorphism  and dissections and explain what they think could be  the possible biological function of KIF6 VUS  to a dissection phenotype. Although I understand that extensive molecular analysis are outside  the scope of the current study a more sound scientific hypothesis  is needed on the presumed function /effect of KIF6 variants on the aortic wall beside the one reported in ‘ Ref 13: Because coronary heart disease (CHD), thoracic aortic dissection, and thoracic aortic aneurysm share risk factors and some aspects of underlying pathophysiology (vessel wall inflammation, common risk factors of hypertension, familial clustering, and smoking) [7, 8], a genetic variant associated with risk for CHD could be also associated with risk for thoracic aortic dissection or aneurysm.

It has recently been reported that carriers of the 719Arg variant in the kinesin-like protein 6 (encoded by KIF6), compared with noncarriers, were at greater risk for CAD in six prospective studies.

The hypothesis on which the current study is based, is that CAD and dissection may have a shared etiology, and  a specific KIF6 variant is a risk factor for both.  The authors need to explain the lack of association between CAD and dissection, as shown in table A1 in the appendix.  In addition TabelA2 indicates that the effect of KIF6 variants is larger in the ascending aorta than in the descending, which is more likely to be affected by KIF 6 related arteriosclerosis. This also needs to be explained.

The study convincingly showed the difference between dissecting and nondissecting TAA patients. To further establish an association with dissections,  the population frequency of the selected KIF6 variants are needed. The authors should add, the population frequencies, and explain the enrichment or lack of it. 

If the KIF6 719Arg variant is very common in the general population this would argue against a strong effect.

In addition it is of interest to report  if and in which cells op the aorta KIF6 is expressed.

A question about family history: please define if this is a family history of aneurysm, or of dissections, if family included only affected first degree relatives or also  affected second degree relatives, and if the family history is reliable, ie the diagnosis in relatives was confirmed by medical records.

Regarding the connective tissue disorders that were excluded; these patients have probably he highest risk of having a dissection. Please explain the definition used for connective tissue disorders how  these diagnoses  were made. This is of importance because- as expected- these disorders have a of connective tissue disorders were made because, as table 1 shows, these patients are more likely to have an dissection, and it would be of interest to find out if the specific KIF6 variant also increases risk in familial, syndromic TAA

Author Response

Reviewer #2

We are extraordinarily grateful to both reviewers. Their extremely thoughtful and insightful comments and suggestions have led us to improve our manuscript, to provide important additional data and clarifications, and to make plans for further extensions of this work in promising directions.

Comments and Suggestions for Authors

The efforts of the authors to broaden our insights into the complex genetic background of aortic aneurysms and dissection are very welcome and much needed. In particular this tool that can predict which patients may develop an dissection has important implication because it allows to select patients for early treatment with better outcome.  In this perspective the current study present a step forwards in the direction of personalized medicine for aortic disease. The manuscript is very well written and easy to read and understand.

We thank the Reviewer for the generous comments, which encourage us in this work.  

As we all know association studies generate questions. Here about the biological effect of identified specific polymorphism in KIF6 on the aortic wall, especially in times were major molecular mechanism of aneurysm are being unraveled.

The authors are therefore encouraged to provide a bit more explanation on the link between the KIF6 polymorphism  and dissections and explain what they think could be  the possible biological function of KIF6 VUS  to a dissection phenotype. Although I understand that extensive molecular analysis are outside  the scope of the current study a more sound scientific hypothesis  is needed on the presumed function /effect of KIF6 variants on the aortic wall beside the one reported in ‘ Ref 13: Because coronary heart disease (CHD), thoracic aortic dissection, and thoracic aortic aneurysm share risk factors and some aspects of underlying pathophysiology (vessel wall inflammation, common risk factors of hypertension, familial clustering, and smoking) [7, 8], a genetic variant associated with risk for CHD could be also associated with risk for thoracic aortic dissection or aneurysm.

Thank you for encouraging us to delve deeper into potential mechanisms for the adverse impact of the KIF6 variant. We have searched the literature and added the following paragraph to the Discussion section of the manuscript.

It is worth examining the biological plausibility of an important clinical impact of KIF6 on aortic biology and clinical prognosis. Firstly, KIF6 has been localized in blood vessels and in the endothelium. (https://pharos.nih.gov/tragets/KIF6)  Furthermore, another study found, in zebrafish, that mutations in KIF6 are related to scoliosis. (Buchan JG, Gray RS, Gansner JM, Alvarado DM, Burgert L, Gitlin JD, Gurnett CA, Goldsmith MI. Kinesin family member 6 (kif6) is necessary for spine development in zebrafish. Dev Dyn. 2014 Dec;243(12):1646-57. doi: 10.1002/dvdy.24208. Epub 2014 Oct 20. PMID: 25283277; PMCID: PMC6207368.) We know that skeletal abnormalities are prominent in syndromic thoracic aortic diseases (eg. Marfan disease). Furthermore, very recently published research from our own team (Prendergast A, Ziganshin BA, Papanikolaou D, Zafar MA, Nicoli S, Mukherjee S, Elefteriades JA. Phenotyping Zebrafish Mutant Models to Assess Candidate Genes Associated with Aortic Aneurysm. Genes (Basel). 2022 Jan 10;13(1):123. doi: 10.3390/genes13010123. PMID: 35052463; PMCID: PMC8775119.) has found that scoliosis appears to be part of the zebrafish phenotype of variants that cause thoracic aortic aneurysm in humans. Thus, there appears to be fundamental biological evidence supporting a possible role of the kinesin family, and KIF6 in particular, in promoting aortic aneurysm disease.

It has recently been reported that carriers of the 719Arg variant in the kinesin-like protein 6 (encoded by KIF6), compared with noncarriers, were at greater risk for CAD in six prospective studies.

The hypothesis on which the current study is based, is that CAD and dissection may have a shared etiology, and  a specific KIF6 variant is a risk factor for both.  The authors need to explain the lack of association between CAD and dissection, as shown in table A1 in the appendix.  In addition TabelA2 indicates that the effect of KIF6 variants is larger in the ascending aorta than in the descending, which is more likely to be affected by KIF 6 related arteriosclerosis. This also needs to be explained.

Thank you for bringing up this important topic. You have motivated us to edit the original second paragraph of the Discussion with addition of the following information, which now addresses these issues directly.

Also, although original studies have associated KIF6 variants with coronary artery disease (CAD), this connection has been questioned as more data became available. (Topal EJ, Damani SB. The KIF6 Collapse. JACC. 2010;56:15640-1566.) In our study (Table A1) we find a lack of association between CAD and dissection. This dissociation between CAD and ascending aortic disease is in keeping with multiple studies from our team (REFS immediately below) showing that ascending aortic aneurysm patients are remarkably free of indications of atherosclerotic disease; in fact, ascending aortic aneurysm patients have less total body vascular calcification, a lower intimal medial thickness (IMT) in the carotid artery, and almost complete protection from myocardial infarction. Also, ascending aortic aneurysm patients have lower LDL than non-aneurysmal patients. These findings seem paradoxical compared to general dogma regarding atherosclerosis until one recognizes that ascending aneurysms are non-calcified, smooth contoured, and non-thrombus containing. This stands in contradistinction to descending and abdominal aortic aneurysms, which are frankly atherosclerotic, with heavily calcified walls, irregular contour, and heavy thrombus burden. These novel perspectives differentiating ascending from descending aortic disease are also completely consonant with the findings in Table A2 that KIF6 has as stronger impact on ascending than on descending aortas.

--Achneck H, Modi B, Shaw C, Rizzo J, Albornoz G, Fusco D, Elefteriades JA. Ascending thoracic aortic aneurysms are associated with decreased systemic atherosclerosis. Chest. 2005;128:1580-1586)

 --Hung A, Zafar M, Mukherjee S, Tranquilli M, Scoutt LM, Elefteriades JA: Carotid intima-media thickness provides evidence that as-cending aortic aneurysm protects against systemic atherosclerosis. Cardiology 2012; 123:71-77.

--Chau K, Elefteriades JA. Ascending thoracic aortic aneurysms protect against myocardial infarctions. Int J Angiol. 2014 Sep;23(3):177-82. doi: 10.1055/s-0034-1382288. PMID: 25317029; PMCID: PMC4169096.

--Weininger G, Ostberg N, Shang M, Zafar M, Ziganshin BA, Liu S, Erben Y, Elefteriades JA. Lipid proviles help to explain protection from systemic atherosclerosis in patients with ascending aortic aneurysm. J Thorac Cardiothorac Surg. 2022;163:e129-32. https://doi.org/10.1016/j.jtcvs.2021.09.031.

The study convincingly showed the difference between dissecting and nondissecting TAA patients. To further establish an association with dissections, the population frequency of the selected KIF6 variants are needed. The authors should add, the population frequencies, and explain the enrichment or lack of it. 

If the KIF6 719Arg variant is very common in the general population this would argue against a strong effect.

We thank the reviewer for this important observation, which has led to our adding the following short paragraph to the Limitations section of the Limitations section of the Discussion, directly employing the Reviewer’s very clear verbiage. Thanks to the Reviewer’s comment, we have also added a Table to the Results section detailing the Whole Exome Sequencing findings in the patients in this study. This is important information, and we appreciate the instructive comment from the reviewer.

As the KIF6 Arg variant is very common in the general population (0.41 frequency in gnomad in a European popullation) (https://gnomad.broadinstitute.org/variant/6-39325078-A-G?dataset=gnomad_r2_1), this argues against a strong effect from this gene alone. We suspect that the adverse effect of KIF6 is minimal unless combined synergistically with other disease-causing aneurysm variants such as those shown in Table 6—in which case KIF6 ARG increases the overall virulence.

Additional Table added:

Pathogenic

Likely pathogenic

VUS (Variants of Unknown Significance)

KIF6 Variant Carriers

5 cases: TGFB3, MYLK, TGFBR2, SMAD3, FBN1

5 cases: TGFBR2, FBN1

70 cases: MIB1, TGFBR3, FBN1, MYH11, COL5A1, ACTA2, MYLK, MYH11, COL1A1, PRKG1, TGFB2, COL3A1, NOTCH1, FLNA, FBN2, SMAD3, COL5A2, TNXB, LOX, HCN4, EMILIN1, ELN, GATA5, BGN

KIF6 Variant Non-Carriers

6 cases: FBN1, Xp deletion, PLOD1, SMAD3

1 case: TNXB

49 cases: COL5A2, TGFBR1, ACTA2, MYH11, FBN1, COL1A1, COL3A1, NOTCH1, FLNA, GATA4, FBN2, COL5A1, MIB1, FLNA, TGFBR2, FBN2, LOX, PRKG1, PKD2

Table 6. Genetic variants found on Whole Exome Sequencing of patients in this study. These are classified here according to the standard American College of Medical Genetics and Genomics categories.

In addition, it is of interest to report if and in which cells op the aorta KIF6 is expressed.

We have added this information within our answer to your first comment (please see above). We have indicated that KIF6 is expressed in blood vessels and in the endothelium.

A question about family history: please define if this is a family history of aneurysm, or of dissections, if family included only affected first degree relatives or also affected second degree relatives, and if the family history is reliable, ie the diagnosis in relatives was confirmed by medical records.

Thank you for urging us to be more precise on these points.

In response, we have indicated in the Methods section that “positive family history” refers specifically to 1st degree relatives. We have also indicated in the Methods section that the senior author determined the family history during the initial clinical History and Physical, reserving the designation “positive family history” for cases with reliable evidence.  

Regarding the connective tissue disorders that were excluded; these patients have probably he highest risk of having a dissection. Please explain the definition used for connective tissue disorders how these diagnoses were made. This is of importance because- as expected- these disorders have a of connective tissue disorders were made because, as table 1 shows, these patients are more likely to have an dissection, and it would be of interest to find out if the specific KIF6 variant also increases risk in familial, syndromic TAA

We apologize for lack of clarity. We did not exclude connective tissue disease. There were 10 patients with connective tissue disease in the study. Theadjusted” rates columns adjusted to connective tissue disease, along with age, sex, family history, bicuspid aortic valve, hypertension, and smoking history. We defined connective tissue disease as Marfan disease, Ehlers-Danlos syndrome, or Loeys-Dietz Syndrome. We have added this definition in the caption of Table 4.